# Eosinophils improve cardiac function after myocardial infarction

Jing Liu[1,2,12], Chongzhe Yang[1,12], Tianxiao Liu[1,2,12], Zhiyong Deng[1,12], Wenqian Fang[1], Xian Zhang[1], Jie Li[1], Qin Huang[1], Conglin Liu[1], Yunzhe Wang[1], Dafeng Yang[1], Galina K. Sukhova[1], Jes S. Lindholt[3,4,5], Axel Diederichsen [4,6], Lars M. Rasmussen[4,7], Dazhu Li[2], Gail Newton[8], Francis W. Luscinskas[8], Lijun Liu[9], Peter Libby [1], Jing Wang [10✉], Junli Guo [1,11✉] & Guo-Ping Shi [1✉]

Clinical studies reveal changes in blood eosinophil counts and eosinophil cationic proteins that may serve as risk factors for human coronary heart diseases. Here we report an increase of blood or heart eosinophil counts in humans and mice after myocardial infarction (MI), mostly in the infarct region. Genetic or inducible depletion of eosinophils exacerbates cardiac dysfunction, cell death, and fibrosis post-MI, with concurrent acute increase of heart and chronic increase of splenic neutrophils and monocytes. Mechanistic studies reveal roles of eosinophil IL4 and cationic protein mEar1 in blocking $H_2O_2$- and hypoxia-induced mouse and human cardiomyocyte death, TGF-β-induced cardiac fibroblast Smad2/3 activation, and TNF-α-induced neutrophil adhesion on the heart endothelial cell monolayer. In vitro-cultured eosinophils from WT mice or recombinant mEar1 protein, but not eosinophils from IL4-deficient mice, effectively correct exacerbated cardiac dysfunctions in eosinophil-deficient ΔdblGATA mice. This study establishes a cardioprotective role of eosinophils in post-MI hearts.

[1] Department of Medicine, Brigham and Women's Hospital and Harvard Medical School, Boston, MA 02115, USA. [2] Laboratory of Cardiovascular Immunology, Institute of Cardiology, Union Hospital, Tongji Medical College, Huazhong University of Science and Technology, Wuhan 430022, China. [3] Department of Cardiothoracic and Vascular Surgery, Odense University Hospital, Odense, Denmark. [4] Elitary Research Centre of personalised medicine in arterial disease (CIMA), Odense University Hospital, Odense, Denmark. [5] Cardiovascular Research Unit, Viborg Hospital, Viborg, Denmark. [6] Department of Cardiology, Odense University Hospital, Odense, Denmark. [7] Department of Clinical Biochemistry and Pharmacology, Odense University Hospital, Odense, Denmark. [8] Department of Pathology, Brigham and Women's Hospital and Harvard Medical School, Boston, MA 02115, USA. [9] Department of Biochemistry and Cancer Biology, College of Medicine and Life Sciences, University of Toledo, Toledo, OH 43614, USA. [10] State Key Laboratory of Medical Molecular Biology, Institute of Basic Medical Sciences, Chinese Academy of Medical Sciences, Department of Pathophysiology, Peking Union Medical College, Beijing 100005, China. [11] Key Laboratory of Emergency and Trauma of Ministry of Education & Research Unit of Island Emergency Medicine, Chinese Academy of Medical Sciences, Hainan Provincial Key Laboratory for Tropical Cardiovascular Diseases Research, The First Affiliated Hospital, Hainan Medical University, Haikou 571199, China. [12]These authors contributed equally: Jing Liu, Chongzhe Yang, Tianxiao Liu, Zhiyong Deng. ✉email: wangjing@ibms.pumc.edu.cn; guojl0511@126.com; gshi@bwh.harvard.edu

osinophils (EOS) develop in the bone marrow under the control of transcription factor GATA-1 and cytokines IL3, IL5, and GM-CSF. EOS cytoplasm have granules that contain cationic proteins such as major basic protein, eosinophilic cationic protein (ECP), eosinophilic peroxidase, cytokines (IL4, IL5, IL10, IL13), and chemokines (CCL-3, CCL-5, CCL-11)[1–3]. EOS have long been considered toxic effector cells that release these granule components upon activation. EOS increase in peripheral blood and accumulate in affected tissues that often accompany hypersensitivity and parasitic diseases[4]. Patients with asthma and other atopic or certain inflammatory diseases have elevated plasma ECP, a biomarker of EOS activation[3,5]. EOS may also contribute to human coronary artery disease (CAD). Blood EOS count increases in patients with acute coronary syndromes (ACS) or after percutaneous coronary intervention (PCI), correlates with CAD prevalence[6], and serves as a biomarker for risk stratification in these patients[7]. In patients with myocardial infarction (MI), blood EOS increase for at least 5 days[8], and plasma ECP peaks within the first 2–3 days of infarct[9]. Plasma ECP associates with coronary atherosclerosis[10] and poor prognosis in patients undergoing PCI[11]. EOS localize in autopsy specimens of cardiac rupture post-MI[12] and atherectomy specimens from patients with in-stent stenosis[13]. Yet, other studies suggest a protective role of EOS in CAD. A low EOS count independently predicts cardiovascular death and correlates negatively with death rates[14]. In the CALIBER study of 775,231 individuals aged 30 or older without CAD at baseline, a strong correlation occurred between low EOS count with heart failure (Hazard Ratio HR: 2.05), unheralded coronary death (HR: 1.94), ventricular arrhythmia/sudden cardiac death, and subarachnoid hemorrhage over 6 months of follow-up. A low EOS count also correlated inversely with peripheral arterial disease (HR: 0.63), although this association became weaker beyond 6 months[15]. This study provides evidence to suggest that increased EOS in peripheral blood and infarcted hearts post-MI play a compensatory and cardioprotective role to reduce cardiomyocyte death, cardiac fibroblast activation and fibrosis, and inflammatory cell adhesion and accumulation.

## Results

**Increased blood EOS counts in patients with previous acute MI.** The Danish Cardiovascular Screening Trial (DANCAVAS) from Odense is a randomized controlled trial of men aged 65–74 years old from southern Denmark[16]. From that study, we consecutively selected 5,864 men who had their blood samples taken and blood EOS counts available (from January 2015 to August 2018). Of these, 345 men had suffered from an acute MI (AMI). Supplementary Table 1 shows the baseline characteristics. Body mass index (BMI), current or former smokers, diabetes, ischemic heart disease, chronic obstructive pulmonary disease, use of low-dose aspirin, statins, loop diuretics, and inhalation therapy associated significantly with high blood EOS counts. Men with previous AMI had significantly higher blood EOS counts than those without AMI ($0.261 \times 10^9$ vs. $0.208 \times 10^9$ EOS/L, $P = 3.3 \times 10E-9$) (Fig. 1a). Multivariate logistic regression analysis suggests that high blood EOS counts posed a significant risk factor of human AMI, and such significance persisted after adjustment for potential confounders (Adjusted odds ratio OR = 2.685 [1.335, 5.401], $P = 0.006$). The BMI group, former or current smoking, and use of statins and low-dose aspirin were also significantly associated with EOS counts, while data from COPD and diabetics did not reveal any significance (Supplementary Table 2). High blood EOS counts associated positively with the Eur-Qol-5D and NYHA classifications. Higher EOS counts are associated with lower mobility. The association with the NYHA classification

group remained significant after adjustment (Supplementary Table 3). In a subgroup of 482 men from our cohort who underwent echocardiography for other research purposes, blood EOS counts correlated significantly and negatively with the ejection fraction (EF, $r = -0.122$, $P = 0.013$). This association significance became weaker but persisted after adjustment of the potential confounders (partial correlation coefficient $r = -0.094$, $P = 0.037$) (Fig. 1b).

**Increased blood and myocardial EOS accumulation post-MI in mice.** Left anterior descending (LAD) coronary artery ligation produced MI in mice. Heart single-cell preparation from 1-day post-MI mice established the FACS gating strategy to detect $CD45^+CD11b^+Siglec-F^+CCR3^+$ EOS (Fig. 1c). Sham-operated mice from wild-type (WT) and EOS-deficient ΔdblGATA mice tested the FACS method specificity (Fig. 1d). As in humans, blood EOS counts increased at 1-day post-MI (Fig. 1e). Results showed a sharp increase in heart EOS 1~3 days post-MI both absolute number per heart and percentage relative to total heart $CD45^+$ cells, many more than those in sham-operated mice. Such an increase declined over time (Fig. 1f). Production of MI in $eoCRE^{+/-}$ $GFP^{+/-}$ mice allowed localization of green fluorescent protein (GFP)-positive EOS in infarcted hearts[17,18]. At 1-day post-MI, GFP antibody immunofluorescent staining detected EOS accumulation mostly in the infarcted region, fewer in the border, and very few in the remote region (Fig. 1g). Results also showed very few EOS in hearts from sham-treated mice (Fig. 1h). Increased heart EOS post-MI suggests that MI induced heart EOS adhesion and chemoattraction. RT-PCR revealed that MI increased the mRNA levels of adhesion molecules (*Icam1, Vcam1, E-selectin*) and chemokines (*Ccr3, Eotaxin, Eotaxin2, Rantes, Mip1a, Mip1b, Mcp1, Mcp2, Mcp3*) in 1-day post-MI heart (Fig. 1i).

**EOS deficiency exacerbates cardiac dysfunctions post-MI.** To test whether increased heart and blood EOS post-MI affect cardiac function and remodeling, we performed sham and MI in 8-week-old male WT and ΔdblGATA mice and measured cardiac functions at 1-month post-MI as outlined in Fig. 2a. As expected, MI production reduced EF and fractional shortening (FS), increased LV end-systolic (LV Vol;s) and end-diastolic (LV Vol; d) volumes, and increased the heart weight-to-body weight ratio (HW/BW) and heart weight-to-tibia length (HW/TL). Deficiency of EOS in ΔdblGATA mice did not improve but further aggravated post-MI cardiac dysfunctions, although the mortality rate did not differ between the groups (Fig. 2b, c). EOS deficiency enlarged infarct size ratio and reduced infarct thickness (Fig. 2d, e), and increased infarct region cell death (TUNEL staining) and collagen deposition (Masson's trichrome staining) (Fig. 2f, g).

Increase of heart EOS post-MI may subsequently increase EOS molecule production. EOS furnish the majority of the Th2 cytokine IL4 in the hearts and accounted for 65% of the IL4 production in the hearts of mice with myocarditis[19–22]. Immunoblot analysis revealed a significant increase of IL4 in the heart from both 1-day and 1-month post-MI WT mice, but greatly reduced IL4 concentrations in those of ΔdblGATA mice (Fig. 2h), indicating that EOS contribute importantly to IL4 production in the heart post-MI. Similarly, mEar1 expression also increased in 1-day post-MI hearts, but ΔdblGATA mice showed a blunted rise (Fig. 2i). A burst of acute inflammation occurs over the first week of MI. Neutrophils are the first innate immune cells recruited to the ischemic myocardium within hours after onset and gradually resolve from day-3 onwards[23–25]. Abrogation of neutrophil influx into the infarcted hearts reduces infarct size and improves survival and cardiac function[26–30]. Pro-inflammatory $Ly6C^{hi}CCR2^+CXCR1^{lo}$ ($CD14^+$ in humans) monocytes

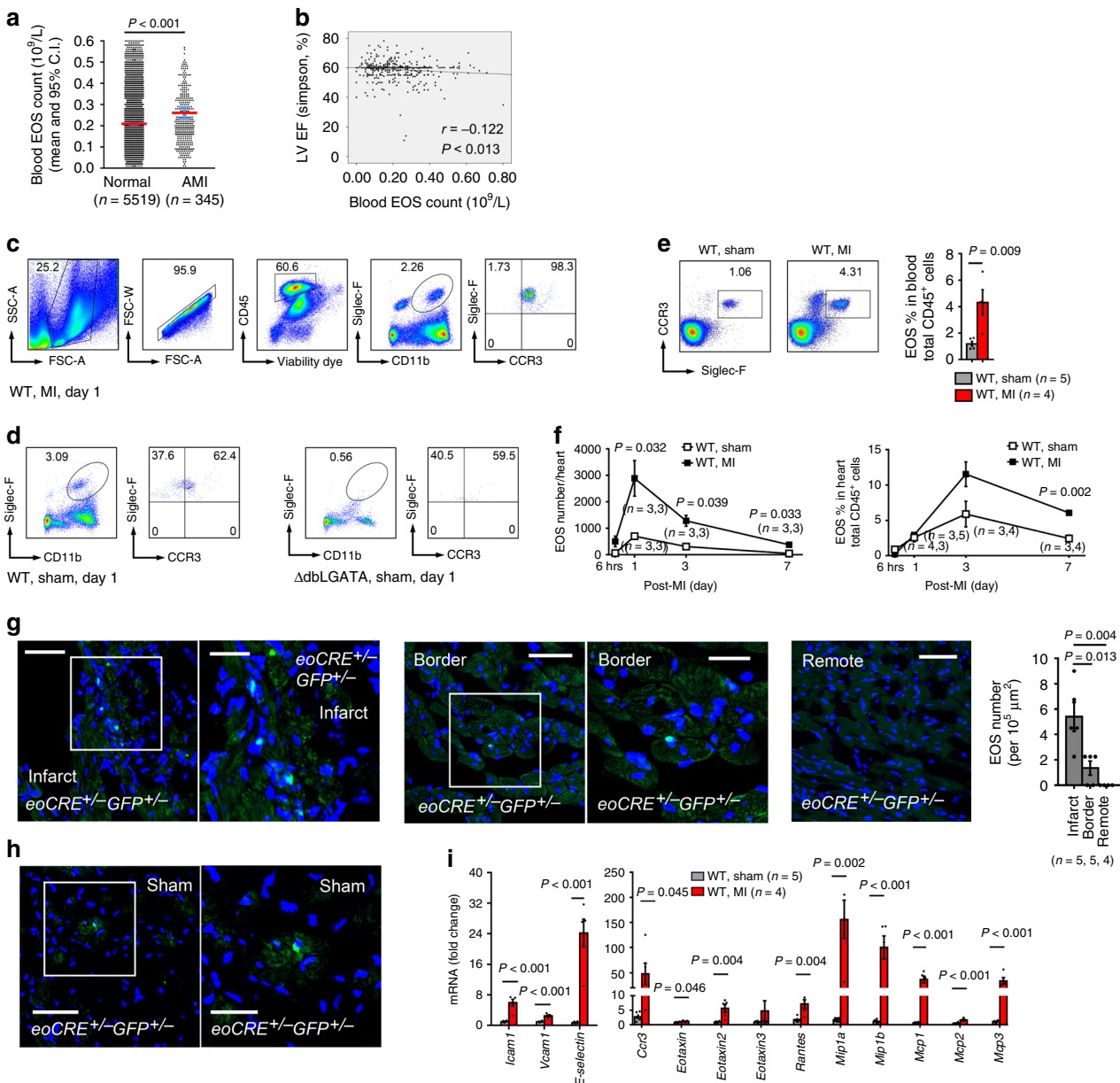

**Fig. 1 Increased human and mouse blood and heart EOS count after MI. a** EOS count per liter blood in men with ($n = 345$) and without ($n = 5,519$) AMI ($P = 3.3 \times 10E-9$). **b**. Spearman correlation between LV EF and blood EOS count among a subgroup of 482 men who had the echocardiography results available ($r = -0.122$, $P = 0.013$). **c**. FACS strategy to analyze heart and blood EOS from WT mice at 1-day post-MI. **d** Representative FACS images of heart EOS from WT sham at 1-day after operation. EOS-deficient ΔdblGATA sham mice served as negative control. **e** FACS analysis of blood EOS in WT sham and MI mice at 1-day after surgery. Representative FACS images are shown to the left. **f**. FACS analysis determined heart EOS count or percentage among total CD45+ cells in WT MI and sham mice at different days after surgery. **g**, **h** Anti-mouse GFP antibody detection of GFP-positive EOS in the infarct, border, and remote regions from *eoCRE*+/−*GFP*+/− mice at 1-day post-MI. Representative images are shown to the left (**g**). Heart sections from sham-operated *eoCRE*+/−*GFP*+/− mice served as experimental control (**h**). Scale: 30 μm, inset: 15 μm. **i**. RT-PCR determined the mRNA levels of EOS relevant chemokines in heart tissues from WT MI and sham mice at 1-day after surgery. Data are mean ± SEM. The numbers of patients (**a**) and mice (**e**–**g**, **i**) in each group, and P values (**a**, **e**–**g**, **i**) are indicated, Student's t-test (**a**), non-parametric Mann-Whitney U test followed by Bonferroni correction (**e**, **f**, **i**), and one-way ANOVA test (**g**).

predominate at 1–3 days post-MI[15,31,32]. Targeting Ly6C[hi] monocyte recruitment to attenuate inflammation enhances myocardial repair[33,34], whereas reparative Ly6C[lo]CCR2−CXCR1[hi] (CD14[dim]CD16+ in humans) monocytes produce VEGF and TGF-β that promote angiogenesis, extracellular matrix protein synthesis, and myocardial healing[23,35]. We monitored the fate of these heart immune cells in WT mice post-MI. At 1-day post-MI, neutrophil and Ly6C[hi] monocytes showed the most abundant expression (Fig. 3a). At 1-day post-MI, ΔdblGATA mouse heart

neutrophils showed higher levels than in WT mice (Fig. 3b). Consistent with this observation, data showed significantly higher expressions of neutrophil chemokines *Cxcl1* and *Cxcl3* in ΔdblGATA hearts than in WT hearts, although other chemokines (*Cxcl2*, *Cxxl5*, *Cxcl7*) did not reach statistical significance (Fig. 3c). At 3 days post-MI, increased Ly6C[hi] monocytes in WT mouse hearts further increased in ΔdblGATA hearts, but the reparative Ly6C[lo] monocytes in ΔdblGATA hearts remained blunted (Fig. 3d). Although the heart monocyte increase was much lower

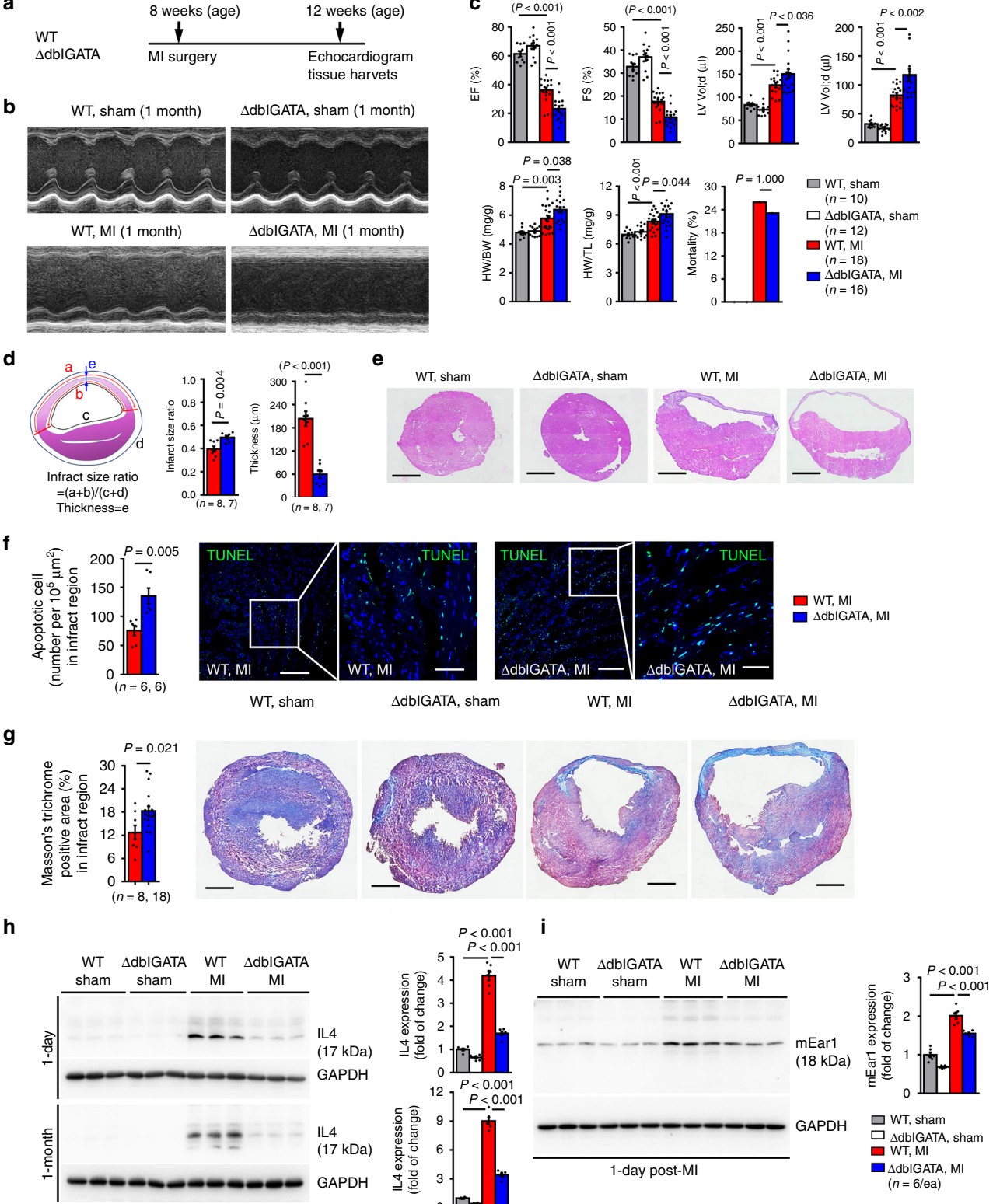

**Fig. 2 EOS-deficiency exacerbates cardiac dysfunction post-MI. a** Mouse surgery strategy. **b** Representative LV M-mode echocardiography images from WT and ΔdblGATA mice at 1-month post-MI or sham. **c** Cardiac functions at 1-month post-MI: EF, FS, LV Vol;d, LV Vol;s, HW/BW ratio, HW/TL ratio, and mortality rate in different groups of mice as indicated. **d** Infarct size ratio, infarct thickness, and calculating formulas. **e** H&E staining for **d** Scale: 1500 μm. **f** TUNEL-positive apoptotic cells in infarct region. Scale: 100 μm, inset: 40 μm. **g** Masson's trichrome staining determined collagen deposition in the infarct region. Scale: 1500 μm. Representative images in **f** and **g** are shown to the right. **h**, **i**. Immunoblots detected IL4 (**h**) and mEar1 (**i**) expression in heart from WT and ΔdblGATA sham and MI mice at 1-day and 1-month post-MI. Data are mean ± SEM. The numbers mice of in each experimental group and *P* values are indicated (**c**, **d**, **f**–**i**), one-way ANOVA test (**c**, **h**, **i**) and non-parametric Mann-Whitney *U* test (**d**, **f**, **g**).

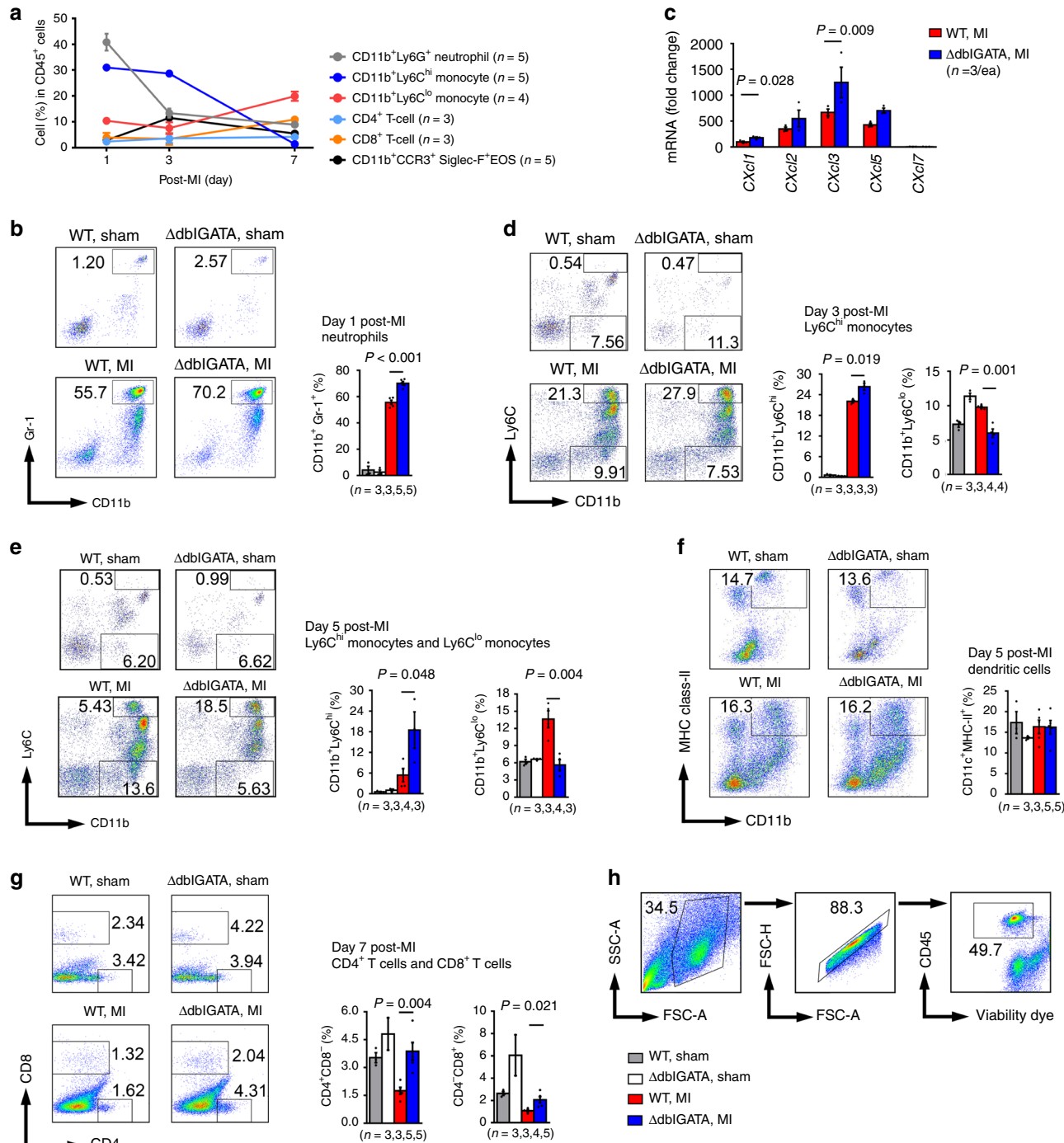

**Fig. 3 EOS deficiency affects heart acute immune cell accumulation post-MI. a** FACS quantification of different immune cells in percentage of total CD45+ cells at different time points post-MI. **b** CD45+CD11b+Gr-1+ neutrophils at 1-day after surgery. **c** RT-PCR determined the mRNA levels of neutrophil relevant chemokines (Cxcl1, Cxcl2, Cxcl3, Cxcl5, Cxcl7) in heart from WT and ΔdblGATA mice at 1-day post-MI. **d, e** CD45+CD11b+Ly6Chi and CD45+CD11b+Ly6Clo monocytes at 3 days (**d**) and 5 days (**e**) after surgery. **f** CD45+CD11c+MHC-II+ dendritic cells at 5 days after surgery. **g** CD45+CD4+CD8- and CD4-CD8+ T cells at 7 days after surgery. **h** FACS gating strategy. Data are mean ± SEM. The numbers of mice in each experimental group and P values are indicated, one-way ANOVA test (**b**, **d**–**g**) and non-parametric Mann–Whitney U test (**c**).

at 5 days post-MI than 3 days post-MI, ΔdblGATA hearts still showed greater numbers of Ly6Chi monocytes than WT hearts, and ΔdblGATA hearts had reduced levels of Ly6Clo monocytes (Fig. 3e). At 5 days and 7 days post-MI, we measured dendritic cells, and CD4+ and CD8+ T cells. Dendritic cell numbers did not differ between WT and ΔdblGATA hearts, but ΔdblGATA hearts contained more CD4+ and CD8+ T cells than WT mouse hearts (Fig. 3f, g). Heart immune cell FACS gating strategy is

shown in Fig. 3h. At 1-month post-MI, we measured splenic inflammatory cells. Splenic neutrophils remained higher in ΔdblGATA mice than in WT mice post-MI (Supplementary Fig. 1a). As in the heart, ΔdblGATA mice had higher levels of post-MI splenic total TNF-α+ inflammatory cells, TNF-α+CD4+, TNF-α+CD8+ T cells, and total CD4+ and CD8+ T cells than WT mice (Supplementary Fig. 1b–e). Yet, data showed significantly reduced splenic CD45+CD4+CD25+Foxp3+ T

regulatory (Treg) cells in ΔdblGATA mice than in WT mice post-MI (Supplementary Fig. 1f). Splenic immune cell FACS gating strategy is shown in Supplementary Fig. 1g. Together, these observations suggest a beneficial role for EOS in MI by diminishing pro-inflammatory cell accumulation and augmenting reparative cells in the myocardium and the spleen.

Female mice behaved differently from males. MI production did not increase the relative EOS percentage in the blood or hearts at 1-day post-MI, although the total numbers of heart CD45[+] immune cells increased (Supplementary Fig. 2a, b). FACS gating strategy is shown in Fig. 1c. As in male mice, post-MI in female WT mice experienced impaired cardiac functions, but EOS deficiency in ΔdblGATA mice did not exacerbate further cardiac dysfunction post-MI (Supplementary Fig. 2c–e). Therefore, the rest of the studies used male mice.

**EOS reduce cardiomyocyte death, fibroblast activation, and neutrophil adhesion.** Myocardial cell death, fibrosis, and leukocyte accumulation are key characteristics of MI and influence subsequent cardiac function. Apoptosis in the infarct and border zones from post-MI hearts occurs predominantly in cardiomyocytes[36–38]. Inhibition of cardiomyocyte apoptosis preserves cardiac function post-MI[39,40]. Although few fibroblasts may undergo apoptosis in post-MI hearts, these cells govern post-MI fibrosis[41]. Tuning of cardiac fibrosis can improve cardiac function post-MI[42]. Blood-borne leukocyte adhesion and accumulation in post-MI heart play an active role in restraining cardiac inflammation[19,43]. In post-MI hearts, most cleaved caspase-3-positive apoptotic cells were myosin heavy chain (MYH)-positive cardiomyocytes that appeared in the infarct and border regions, but not in the remote area (Fig. 4a). In contrast, few α-SMA-positive fibroblasts stained positive for cleaved caspase-3 by immunofluorescent double-staining (Fig. 4b). To test a direct role of EOS on cardiomyocyte death, we treated adult mouse cardiomyocytes with $H_2O_2$ to induce cell death with and without EOS lysate preparation for 4 hrs. WT EOS significantly blocked $H_2O_2$-induced cardiomyocyte death (Fig. 4c). In addition, WT EOS also reduced $H_2O_2$-induced death of inflammatory cells, including neutrophils, macrophages, total lymphocytes, and CD4[+] and CD8[+] T cells (Supplementary Fig. 3a–f). Supplementary Fig. 4a–e present the corresponding FACS gating strategies. WT EOS also blocked TGF-β-induced Smad2 and Smad3 activation in mouse cardiac fibroblasts in a dose-dependent manner (Fig. 4d, e). TNF-α induced ICAM-1 and VCAM-1 expression in monolayers of mouse heart endothelial cells (MHECs), as determined by FACS analysis (Fig. 4f). Expression of these adhesion molecules enhanced neutrophil adhesion. Pretreatment of the inflamed MHEC with WT EOS significantly blocked neutrophil adhesion. In contrast, EOS from $Il4^{-/-}$ mice or WT EOS pre-treated with an anti-mEar1 antibody failed to alter neutrophil adhesion. EOS from $Il13^{-/-}$ mice acted similarly to WT EOS in blocking TNF-α-induced neutrophil adhesion to MHEC monolayers (Fig. 4g). Figure 4h shows the FACS gating strategy of Fig. 4c. Therefore, EOS-derived IL4 and mEar1, but not IL13, are essential for neutrophil adhesion. These observations suggest that EOS mitigate post-MI cardiac functions by protecting cardiomyocytes from apoptosis, suppressing (myo) fibroblast fibrosis, and inhibiting leukocyte adhesion and accumulation, in addition to affecting the myocardium and systemic inflammatory cell profiles.

**EOS-derived IL4 and mEar1 contribute to EOS cardioprotective activity post-MI.** EOS produce IL4, IL10, and IL13 and secrete cationic proteins, such as ECP[1–3]. Immunoblot analysis showed increased IL4 and mEar1 in hearts from WT mice at 1-day post-MI while no such increase was seen in hearts from ΔdblGATA mice (Fig. 2h, i). Our MHEC-EOS-neutrophil co-

culture experiment demonstrated a role for EOS-derived IL4 and mEar1 in blocking neutrophil adhesion (Fig. 4g). Therefore, we tested whether EOS-derived IL4 and mEar1 play a direct role in protecting the heart from MI injury by treating ΔdblGATA mice with WT EOS, $Il4^{-/-}$ EOS, PBS (phosphate-buffered saline), and recombinant mEar1. ΔdblGATA mice received two doses of EOS adoptive transfer followed by MI production. Mice also received mEar1 or PBS via a subcutaneously implanted minipump (Fig. 5a). FACS analysis of EOS prepared from CD45.1 mice ensured the purity of in vitro-prepared EOS (Supplementary Fig. 5a, b), donor EOS targeting, and their fate in recipient mouse hearts and blood (Supplementary Fig. 5c, d). FACS gating strategy for Supplementary Fig. 5c, d is shown in Fig. 3h. Donor EOS from WT mice and recombinant mEar1, but not EOS from $Il4^{-/-}$ mice, reversed cardiac dysfunction, infarct size enlargement, infarct thinning, myocardial collagen deposition (Fig. 5b–e), and the expression of myofibroblast markers α-SMA and CD90 in the infarct region, although only WT EOS reduced the expression of these myofibroblast markers in the border region in ΔdblGATA-recipient mice (Supplementary Fig. 6a–d). These observations suggest that EOS-derived IL4 and mEar1 contributed to the EOS cardioprotective function post-MI. Used at the same dose (1 μg/mouse/day) and treatment, control plasma protein mouse serum albumin (MSA) did not display any difference from PBS in post-MI cardiac dysfunctions in ΔdblGATA recipient mice (Supplementary Fig. 7a–c). The same dose of mEar1 did not affect cardiac dysfunction post-MI in WT mice (Supplementary Fig. 8a–c), suggesting that mEar1 corrected only the exacerbated portion of cardiac dysfunctions from EOS deficiency in ΔdblGATA mice, or that showing significant effects on WT mice might require a much higher dose of mEar1 than 1 μg/mouse/day.

To $H_2O_2$-induced death of adult mouse cardiomyocytes, EOS from WT, $Il10^{-/-}$, and $Il13^{-/-}$ mice, but not EOS from $Il4^{-/-}$ mice blocked cardiomyocyte death. WT EOS lost this activity when EOS were treated with an anti-mEar1 antibody (Fig. 5f), suggesting a role of EOS-derived IL4 and mEar1 in protecting cardiomyocyte from $H_2O_2$-induced death. Cardiomyocyte apoptosis FACS gating strategy is shown in Fig. 4h. This hypothesis was further tested by treating $H_2O_2$-induced cardiomyocytes with and without recombinant IL4 and mEar1, both of which elevated cardiomyocyte Bcl-2 expression in a dose-dependent manner (Fig. 5g, h). In contrast, only mEar1, but not recombinant IL4 suppressed TGF-β-induced Smad2/3 activation in mouse cardiac fibroblasts (Fig. 5i, j). Under the same experimental conditions, however, only EOS from WT and $Il10^{-/-}$ mice, but not those from $Il4^{-/-}$ and $Il13^{-/-}$ mice blocked TGF-β-induced Smad2/3 activation in mouse cardiac fibroblasts (Fig. 5j), supporting an indirect role of EOS-derived IL4 in regulating mouse cardiac fibroblast Smad2/3 activation. To test this possibility, we examined the mEar1 expression in EOS from WT and $Il4^{-/-}$ mice and found a significant reduction of mEar1 expression in EOS from $Il4^{-/-}$ mice (Supplementary Fig. 9a). Therefore, EOS from $Il4^{-/-}$ mice failed to block Smad2/3 activation in cardiac fibroblasts likely because of their reduced expression of mEar1 and possibly other untested molecules. Indeed, plasma IL4 levels did not differ between WT and ΔdblGATA mice at 1-month post-MI (Fig. 5k). In hearts from WT and ΔdblGATA mice, mEar1 expression lowered at 1-month post-MI compared with hearts from sham mice. Subcutaneous minipump-mediated delivery of mEar1 significantly increased heart mEar1 levels (Supplementary Fig. 9b). Increase of heart mEar1 levels in ΔdblGATA mice may explain why administration of mEar1 at this dose improved post-MI cardiac dysfunctions in these mice (Fig. 5a–e). Yielding significant beneficial outcomes in

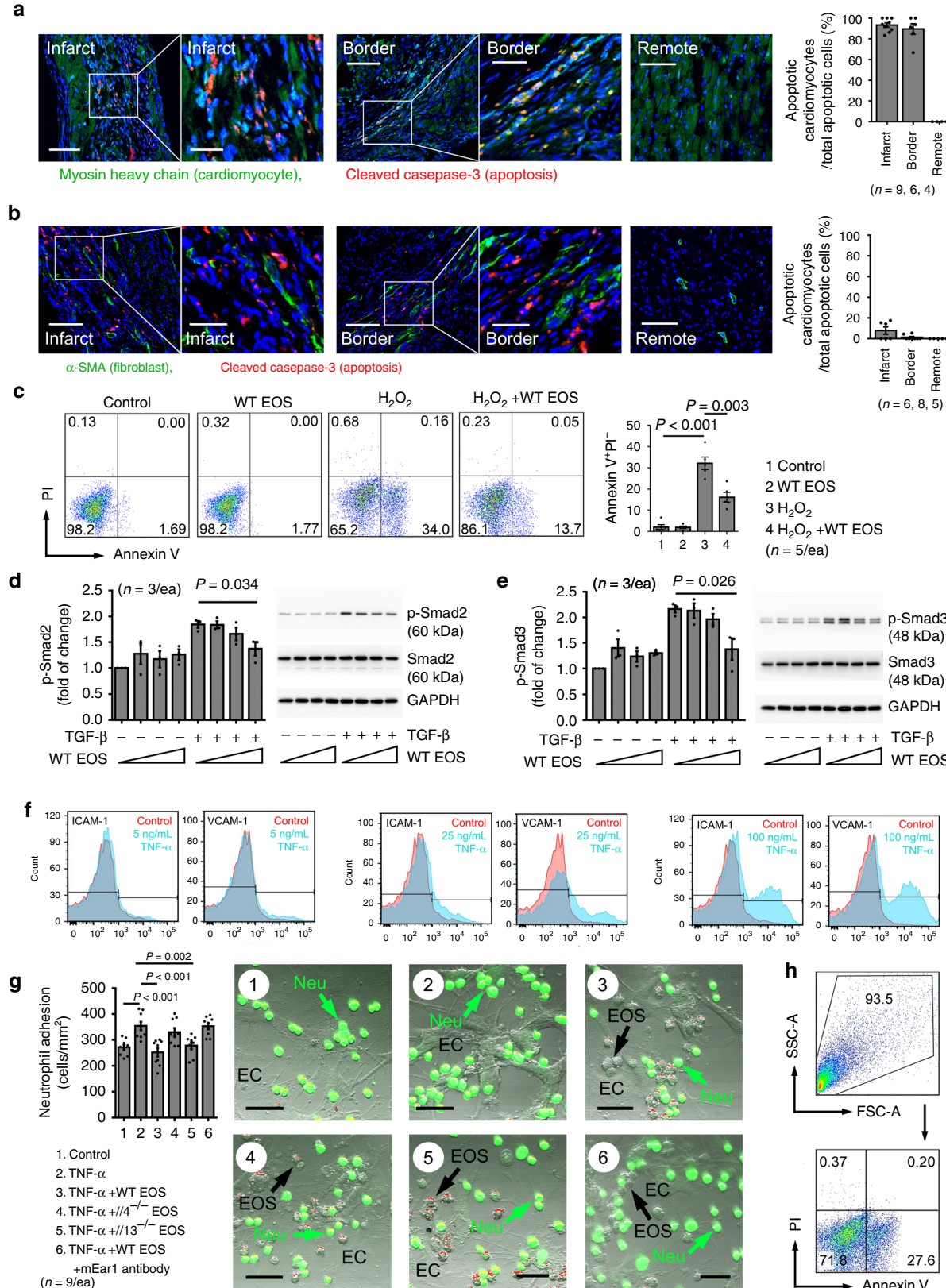

WT mice may require a much higher dose of mEar1 (Supplementary Fig. 8a–c). Together, these results identify mEar1 as an essential EOS molecule responsible for EOS-mediated protection of cardiac function post-MI.

**Diphtheria toxin-induced EOS depletion also worsens cardiac dysfunction post-MI.** Genetic deficiency of EOS in ΔdblGATA mice may affect the development or function of other immune cells that may indirectly contribute to exacerbated cardiac dysfunction in these mice (Fig. 2a–g). To address this possibility, we

**Fig. 4 EOS activities in cardiac cell death, fibrosis, and cell adhesion. a, b** Immunofluorescent staining detected cleaved caspase 3-positive cardiac myosin heavy chain (MYH)-positive cardiomyocytes (**a**) and α-SMA-positive fibroblasts (**b**) in infarct, border, and remote regions of WT mice at 1-month post-MI. Scale: 100 μm, inset: 40 μm. **c** FACS detection of Annexin V$^+$PI$^-$ (propidium iodide) early apoptotic cardiomyocytes after cells were treated with and without $H_2O_2$ (100 μM) and EOS lysate (equivalent to $10^6$ EOS/ml) from WT mice. Representative images in **a–c** are shown to the left. **d, e** Immunoblot detection of p-Smad2/3, total Smad2/3, and GAPDH in fibroblasts treated with 10 ng/ml TGF-β and different concentrations of EOS lysate (equivalent to $1 \times 10^5$, $5 \times 10^5$, $1 \times 10^6$ EOS per ml) for 30 min. Representative immunoblots are shown to the right. **f** FACS detection of ICAM-1 and VCAM-1 expression from MHECs after treatment with and without (control) different concentrations of TNF-α (5, 25, and 100 ng/ml). **g** Adhesion of 5-(and 6)-carboxyfluorescein diacetate succinimidyl ester (CFSE)-labeled neutrophils on TNF-α (100 ng/ml)-treated MHECs after pretreatment with different types of EOS as indicated. Scale: 50 μm. **h** FACS gating strategy for panel **c**. The numbers of mice (**a, b**), the numbers of experiments (**c–e, g**), and *P* values are indicated, one-way ANOVA test.

produced MI in iPHIL mice, in which intraperitoneal injection of diphtheria toxin (DTX) selectively depleted EOS without affecting other immune cells[17]. As in the ΔdblGATA mice, DTX-induced EOS depletion also exacerbated cardiac dysfunctions, along with larger infarct size ratio and thinner infarct thickness compared to untreated mice (Fig. 6a–d). FACS analysis confirmed effective EOS depletion in the spleens and hearts at 1-day post-MI and sham mice after DTX treatment (Supplementary Fig. 10a). As in the dblGATA mice, inducible depletion of EOS also increased heart neutrophils at 1-day post-MI, increased heart Ly6C$^{hi}$ monocytes and reduced heart Ly6C$^{lo}$ monocytes at 3 days post-MI, and increased heart CD8$^+$ T cells at 7 days post-MI (Supplementary Fig. 10b–d). FACS gating strategy is shown in Fig. 3h. At 1 month post-MI, only neutrophils and Ly6C$^{hi}$ monocytes in the spleen remained higher in DTX-treated mice than in the control mice (Supplementary Fig. 11a, b). Other cells, including total TNF-α$^+$ cells, TNF-α$^+$CD4$^+$ T cells, TNF-α$^+$CD8$^+$ T cells, and total CD4$^+$ and CD8$^+$ T cells, Treg cells, and EOS did not differ between DTX-treated and control mice (Supplementary Fig. 11b–h). Splenic immune cell FACS gating strategy is shown in Supplementary Fig. 1g.

To explain why splenic Treg cells were reduced in ΔdblGATA mice at 1-month post-MI (Supplementary Fig. 1f), but not in DTX-induced EOS-depleted mice (Supplementary Fig. 11g), we measured plasma IL2 and TGF-β levels in WT and ΔdblGATA mice at 1-month post-MI. Both IL2 and TGF-β did not differ between WT and ΔdblGATA, sham or 1-month post-MI (Supplementary Fig. 12a, b), suggesting that reduced splenic Treg cells in ΔdblGATA mice 1-month post-MI was not because of a Treg differentiation defect. During the IL2/TGF-β-induced Treg differentiation, the co-culture of WT EOS reduced Treg production (Supplementary Fig. 12c–e). Therefore, the mechanism of reduced splenic Treg cells in ΔdblGATA mice but not in DTX-treated iPHIL remains unknown.

As in the ΔdblGATA mice, DTX-induced EOS depletion in iPHIL mice also increased myocardial cell death, as determined by TUNEL staining (Fig. 6e). In contrast to the ΔdblGATA mice, which showed significantly higher collagen deposition in the infarct region than in WT mice at 1 month post-MI (Fig. 2g), induced EOS depletion did not affect myocardial collagen deposition at 1 month post-MI (Fig. 6f). Comparable EOS contents at this time point when collagen synthesis was dominant may have led to this observation (Supplementary Fig. 11h). We tested this hypothesis by examining the mRNA levels of both collagen-I and collagen-III at different days post-MI. The synthesis of these collagens increased after 7 days post-MI and reached to the highest level at 28 days post-MI (Supplementary Fig. 13a, b).

**Human EOS block human cardiomyocyte death and cardiac fibroblast activation.** Our mouse studies suggest that increased blood and heart EOS in post-MI hearts play a cardioprotective role. Increased blood EOS counts in those AMI patients may also exert a compensatory and cardioprotective function (Fig. 1a). To test whether human EOS acted the same as mouse EOS on human cardiac cells, we induced human cardiomyocyte apoptosis by hypoxia. Human EOS dose-dependently blocked hypoxia-induced human cardiomyocyte apoptosis, as determined by TUNEL staining (Fig. 6g). Similar to mouse EOS, human EOS also dose-dependently blocked TGF-β-induced Smad2/3 activation in human cardiac fibroblasts (Fig. 6h, i).

## Discussion

This study establishes a cardioprotective role for EOS in ischemia-injured mouse hearts. Both genetically EOS-deficient ΔdblGATA mice and DTX-induced EOS-depleted iPHIL mice yielded the same conclusion. While we revealed much higher blood EOS counts in patients with AMI than in those without AMI, MI production in mice also increased blood and heart EOS accumulation. Results from this study support the hypothesis that an increase of heart and blood EOS post-MI represents a compensatory mechanism to protect the heart from ischemia injury. Although additional mechanisms may participate, this study demonstrated beneficial activities of EOS in protecting cardiomyocytes from $H_2O_2$- or hypoxia-induced cell death, which is one of the major events in ischemic hearts[36–38], reducing TGF-β-induced cardiac fibroblast Smad2/3 activation and collagen synthesis that govern cardiac function[41,42], and blocking the adhesion of neutrophils and possible other untested inflammatory cells that may reduce cardiac infiltration of these inflammatory cells, which remain essential to the cardiac reparative mechanisms[19,43]. Further mechanistic studies revealed an important role for EOS-derived IL4 and cationic protein mEar1 in mediating the aforementioned beneficial EOS functions. Consistent with increased blood EOS counts, patients with non-ST-elevation ACS or stable angina also contain higher plasma ECP levels than healthy patients[10], indicating enhanced EOS activation. In acute MI patients, serum ECP levels peak within the first 2~3 days of infarct[9], but some have proposed that ECP can augment endothelial adhesion-molecule expression and facilitate monocyte recruitment to the myocardium[5,20]. Yet, the present experimental results support a cardioprotective role of mEar1 in mouse hearts post-MI, consistent with prior studies that ECP inhibits T-cell responses to antigens, blocks immunoglobulin synthesis, and modulates fibroblast activity[5,21].

This study did not explore many other potentially important EOS cytokines and cationic proteins, which might also participate in reducing post-MI cardiac cell death, fibrosis, and inflammatory cell recruitment. Although our data demonstrated a direct role of mEar1, the human ECP ortholog, in reducing heart ischemia-induced injuries on all tested key cardiac cells (cardiomyocytes, cardiac fibroblasts, and MHECs), EOS cytokines may display both direct and indirect roles. This study reported a direct role of IL4 in reducing cardiomyocyte death, and also an indirect role of IL4 in cardiac fibroblast fibrotic signaling by affecting the expression of mEar1 and other untested molecules. Earlier studies showed that IL4 is the major Th2 cytokine of EOS and human blood EOS

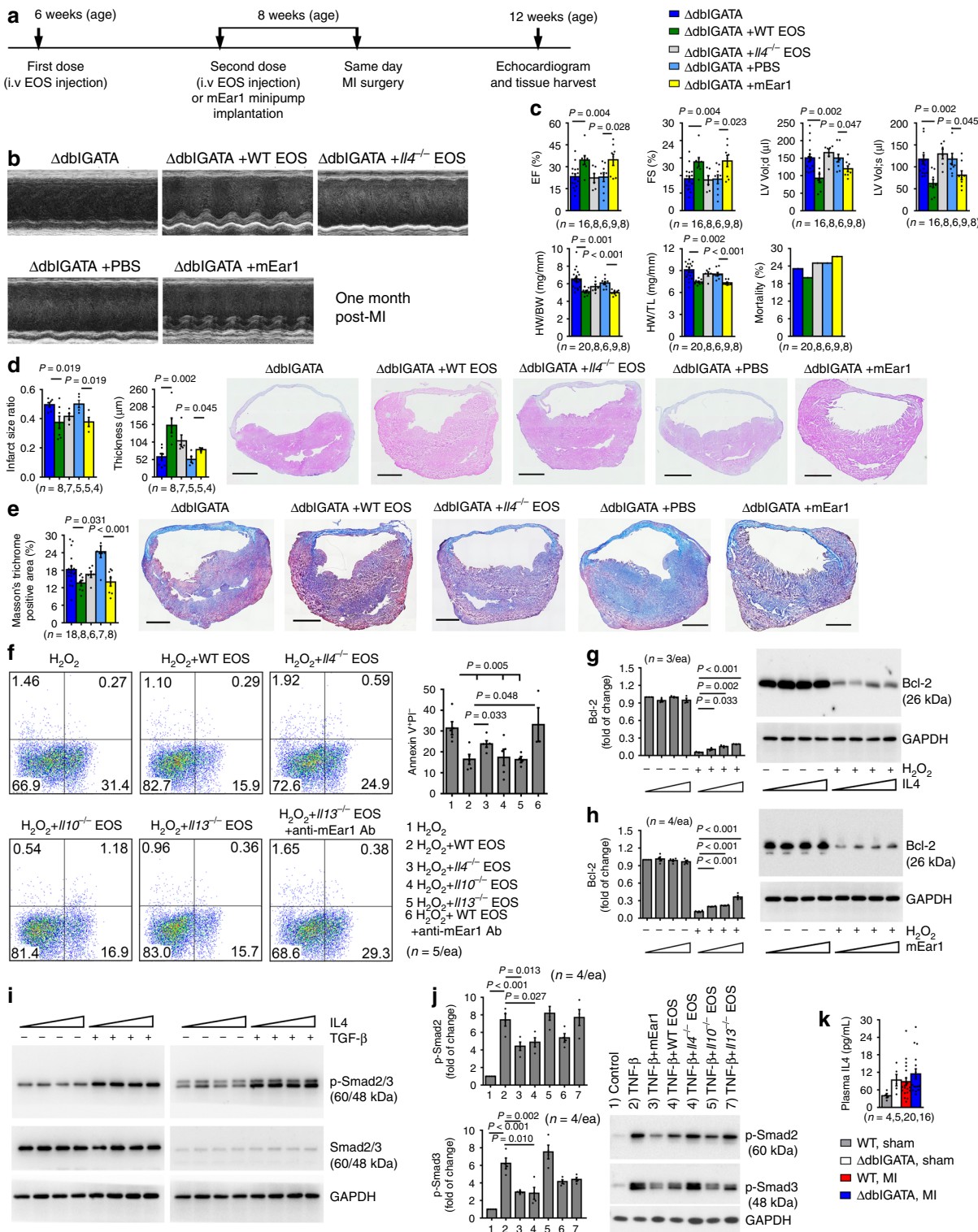

produce 2–4 ng IL4 per $10^6$ cells in 3 h[22,44]. Under inflammatory conditions, human EOS showed a greater than 50–200 fold increase of IL4 production[45]. For example, EOS accounted for 65% of the IL4 production in mouse heart with myocarditis[46]. EOS may still produce the majority of IL4 in the ischemicaly injured mouse hearts. Observations, such as the greatly reduced augmentation in IL4 in ΔdblGATA mice post MI, support this hypothesis. Yet, Th2 and NKT cells in post-MI hearts may also contribute to IL4 production in this context[47–49]. IL4 from these inflammatory cells may exert actions similar to that of IL4 from

EOS in repairing heart from post-MI injury, a hypothesis that this study did not test.

Of note, this study left several unresolved questions that also merit further investigation. First, as discussed, we only have indirect evidence to support the role of EOS-derived IL4 in cardiac fibrosis by regulating EOS mEar1 expression, although IL4 did block $H_2O_2$-induced cardiomyocyte death, and EOS-deficient ΔdblGATA mice at 1-day and 1-month post-MI had significantly reduced heart IL4 levels. Other EOS cytokines may act in similar ways. Further, each cytokine may also act differently from the

**Fig. 5 EOS-derived IL4 and mEar1 protect cardiac function post-MI. a.** Mouse surgery and treatment strategy. **b.** Representative LV M-mode echocardiography images at 1-month post-MI. **c.** Cardiac functions at 1-month post-MI: EF, FS, LV Vol;d, LV Vol;s, HW/BW, HW/TL, and mortality rates of different mice as indicated. **d, e.** Infarct size ratio, thickness, and Masson's trichrome-positive areas of hearts from different groups of mice as indicated at 1-month post-MI. Representative H&E and Masson's trichrome staining images are shown to the right. Scale bars: 1500 μm. **f.** FACS detection of Annexin V$^+$ PI$^-$ early apoptotic cardiomyocytes after cells were exposed to 100 μM $H_2O_2$ with or without different types of EOS lysate (equivalent to $10^6$ EOS/ml) or together with anti-mEar1 antibody. Representative FACS images are shown to the left. FASC gating strategy is shown in Fig. 4h. **g, h.** Immunoblot detection of Bcl-2 and GAPDH in mouse cardiomyocytes treated with IL4 (0, 10, 50, 100 ng/ml) (**g**) or mEar1 (0, 10, 100, 1000 ng/ml) (**h**) with or without $H_2O_2$ (100 μM). Representative blots are shown to the right. **i.** Immunoblots of p-Smad2/3, total Smad2/3, and GAPDH in fibroblasts treated with IL4 (0, 10, 50,100 ng/ml) and then with and without TGF-β (10 ng/ml) for 30 min. Data represent 4 independent experiments. **j.** Immunoblots of p-Smad2/3 and GAPDH in fibroblasts treated with different types of EOS lysates (equivalent to $10^6$ EOS/ml) and TGF-β (10 ng/ml) for 30 min. Representative blots are shown to the left. **k.** Sham and 1-month post-MI mouse plasma IL4 levels from different groups as indicated. Data are mean ± SEM. The numbers of mice (**c–e**, **k**), the numbers of independent experiments (**f–j**), and P values are indicated, one-way ANOVA test.

other. For example, $Il13^{-/-}$ EOS acted the same as WT EOS in blocking neutrophil adhesion on the MHEC monolayer and $H_2O_2$-induced cardiomyocyte death, and in inhibiting Smad3 activation, but failed to block Smad-2 activation. Second, our presented cardioprotective role of EOS applied only to male mice. In female mice, EOS deficiency did not show any phenotypic difference post-MI from WT control mice. We did not further test the mechanisms behind these observations and we do not know whether this holds true in humans. Our cohort will not answer this question as the DANCAVAS trial contains only men. Third, as discussed, spleens from ΔdblGATA mice when compared with WT control mice at 1-month post-MI showed reduced levels of Treg cells. Such a reduction did not occur in spleens from DTX-treated iPHIL mice *versus* untreated control iPHIL mice. Reduced splenic Treg cells in ΔdblGATA mice may not directly result from EOS deficiency but possibly from changes of other inflammatory cells due to EOS deficiency because the co-culture of EOS did not promote, but rather reduced the Treg cell differentiation. Fourth, EOS accumulation in the heart post-MI was relatively low compared with many other inflammatory cells. At 1 day post-MI, there were about 3000 EOS per heart, many fewer than neutrophils and Ly6C$^{hi}$ and Ly6C$^{lo}$ monocytes. At 3 days post-MI, heart EOS counts reached to the level of neutrophils, but still about one-third of Ly6C$^{hi}$ monocytes. Results from this study suggest that EOS in the heart post-MI affects the accumulation of other inflammatory cells. It is also possible that the accumulation of pro-inflammatory and reparative inflammatory cells control EOS accumulation, a hypothesis remains untested.

As summarized in Fig. 7, this study suggests a beneficial role of EOS in infarcting hearts by producing IL4, mEar1 (human ECP ortholog), and possibly other untested molecules, mitigating the cardiac inflammatory cell profile, limiting cardiomyocyte apoptosis, modulating fibroblast activity, and regulating post-MI heart inflammatory cell adhesion and infiltration. These results shed mechanistic light on the modulation of inflammation during the healing of myocardial ischemic injury.

## Methods

**Humans.** The human population included men attending the Danish Cardiovascular Screening Trial (DANCAVAS) in Odense. We consecutively selected 5,864 men who had blood EOS counts available. The DANCAVAS trial, which has been described in detail elsewhere[16], is a population-based, randomized, and clinically-controlled screening trial of men aged 65–74. No exclusion criteria were used. One-third was invited for cardiovascular screening examinations including a CT scan at one of the four locations, among which 62.4% men attended. The screening includes a low-dose, noncontrast computerized tomography scan to detect coronary artery calcification and aortic and iliac aneurysms; brachial and ankle blood pressure index to detect peripheral arterial disease and hypertension; a telemetric assessment of heart rhythm; and a measurement of blood cholesterol and glucose levels. For the purpose of this study, leucocyte count, including EOS, was also performed. At attendance of screening, each man was first given the informed consent, then a medical history was obtained including previous acute myocardial

infarction (AMI), coronary revascularization, chronic obstructive pulmonary disease (COPD), medication history, and symptoms. In case of dyspnea, NYHA classification was performed. Up-to-date cardiovascular preventive treatment was recommended in case of subclinical cardiovascular disease. Use of anonymized patient information from the DANCAVAS was approved by the Human Investigation Review Committee at the Brigham and Women's Hospital, Boston, MA, USA (protocol #2010P001930) with a waiver of informed consent since the study did not involve patient contact or enrollment.

**Mice.** Male or female $Il4^{-/-}$ (002253) and $Il10^{-/-}$ (002251), 45.1 transgenic mice (002014), C57BL/6 mice (000664), male and female ΔdblGATA (005653), and Balb/c (000651) mice aged 7–8 weeks were purchased from the Jackson laboratory (Bar Harbor, ME). We previously reported the $Il13^{-/-}$ mice[50]. eoCRE and EOS-less iPHIL mice were described previously[17]. eoCRE mice were crossed with a (flox-stop-flox)-GFP reporter strain (B6.Cg-Gt(ROSA)26Sortm6(CAG-ZsGreen1)Hze/J, 007906) purchased from the Jackson laboratory. All mice were housed in a pathogen-free facility on a 12 light/12 dark cycle at temperatures of 65–75 °F with 40–60% humidity. All animal procedures conformed to the Guide for the Care and Use of Laboratory Animals published by the US National Institutes of Health, and were approved by the Brigham and Women's Hospital Standing Committee on Animals (protocol #2016N000442).

**MI production.** Mouse MI was induced under anesthesia with 1.5% isoflurane and a volume-controlled ventilator (Harvard Apparatus, Holliston, MA) as we described previously[51]. Briefly, after a left thoracotomy, the heart was exposed and MI was produced by permanent ligation of the left anterior descending (LAD) artery through a 7-0 silk suture (Ethicon, Somerville, NJ). Bleaching of the distal myocardium verified ischemia. Sham-operated mice underwent the same procedure without LAD ligation.

To assess the role of EOS in MI, 8-week-old male or female EOS genetic-deficient ΔdblGATA mice and Balb/c WT control mice were used and divided for MI production or sham operation. Mice were sacrificed at different days (1, 3, 7, 30) post-MI for further studies. Before harvest, an echocardiogram was performed and body weight was obtained. At harvest, mouse tibia length was measured. Hearts and spleens were collected for immunostaining or to prepare single-cell suspension for FACS analysis. To test the role of EOS-derived IL4 and EOS cationic protein mEar1 in mouse MI, we performed an adoptive transfer of EOS from WT or $Il4^{-/-}$ mice to ΔdblGATA-recipient mice by intravenous (i.v.) injection at 10 days before MI and 30 min before the surgery at a dosage of $10^7$ donor EOS per administration, or gave phosphate-buffered saline (PBS) or mEar1 (1 μg per mouse per day for 30 days) using a subcutaneous implanted Alzet minipump. EOS depletion in iPHIL mice was induced by giving mice intraperitoneal injection of diphtheria toxin (DTX, 0564, Sigma-Aldrich, St. Lois, MO) at 300 ng per mouse for 3 days before MI production, and twice a week for the first 2 weeks post-MI.

**Echocardiography.** A Vevo 3100 high-frequency micro-ultrasound imaging system (VisualSonics, Toronto, Canada) with a 70-MHz transducer was used to measure mouse cardiac function. Mice were conscious and properly positioned and restrained on a board during examination. Two-dimensional echocardiographic views of the mid-ventricular short axis and parasternal long axes M-mode was obtained at baseline and at 30 days post-MI. M-mode tracings at mid-papillary muscle level were recorded to measure left ventricle (LV) wall thickness, end-diastolic diameter (LV EDD), and end-systolic diameter (LV ESD). Percentage of fraction shortening (%FS) and ejection fraction (%EF) and other cardiac functions were calculated by the built-in software package. Vevo Lab v3.2.0 software was used for echocardiography analysis.

**Heart total single-cell preparation and flow cytometry.** The heart was perfused with 20 ml of cold PBS and then removed from the mouse. Heart tissue was minced into small pieces and digested in a 0.1% collagenase B (LS004177, Worthington Biochemical Co, Lakewood, NJ) dissolved in a HEPES buffer for 30 min in a 37 °C

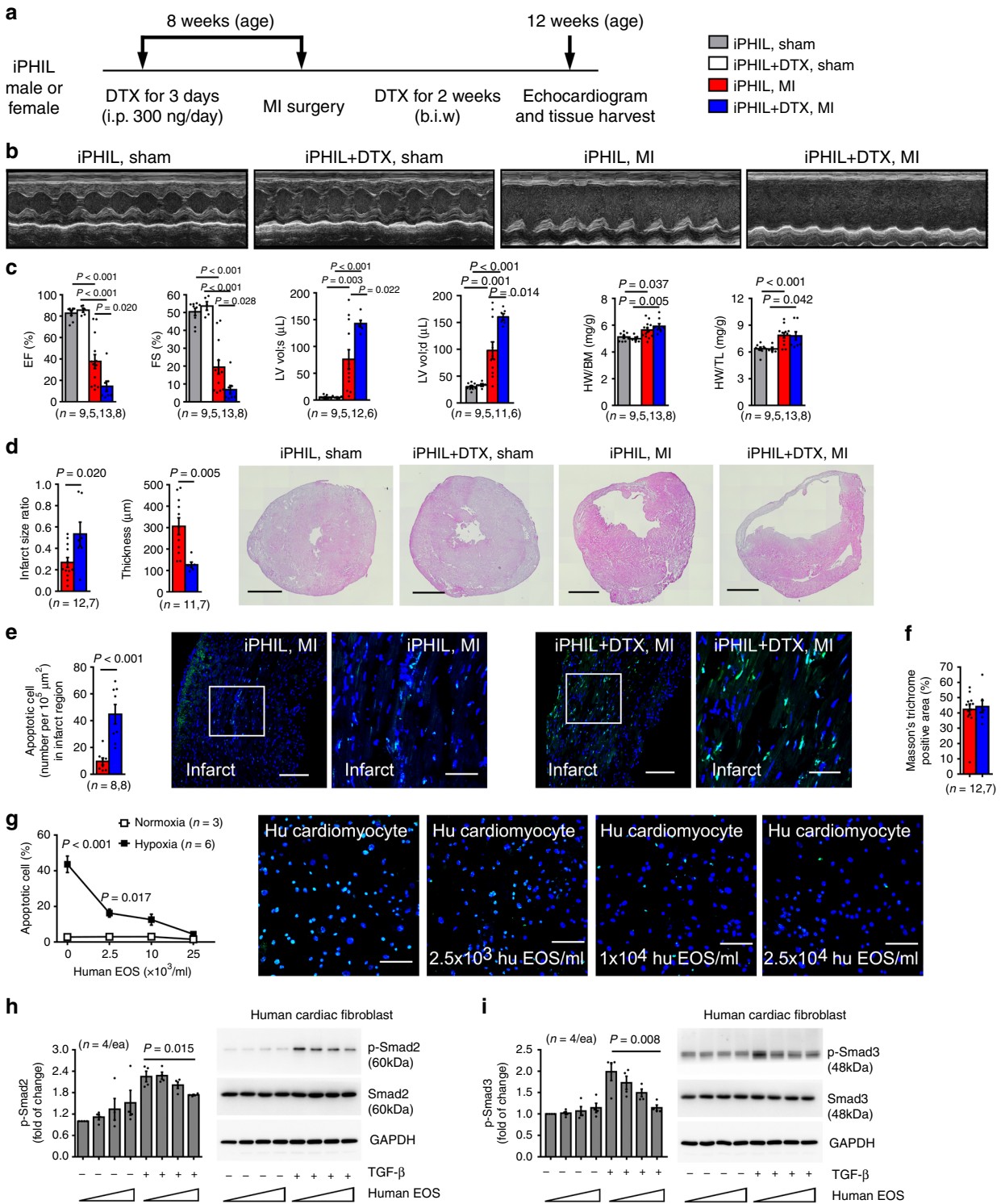

**Fig. 6 DTX-induced EOS depletion exacerbates cardiac functions post-MI. a** Mouse surgery and treatment strategy. **b** Representative LV M-mode echocardiography images at 1-month post-MI from different groups of mice. **c** EF, FS, LV Vol;d, LV Vol;s, HW/BW, and HW/TL in different groups of mice. **d–f** Infarct size ratio and thickness (**d**), infarct TUNEL-positive apoptotic cells (**e**), and infarct region Masson's trichrome-positive areas (**f**) in hearts from different groups of mice as indicated. Representative H&E staining (Scale: 1500 μm) and TUNEL staining (Scale: 100 μm, inset: 30 μm) images are shown to the right. **g** TUNEL staining of human cardiomyocytes treated with different doses of human EOS lysates (equivalent to $2.5 \times 10^3$, $1 \times 10^4$, $2.5 \times 10^4$ EOS/ml) under hypoxia and normoxia conditions for 36 hrs. Representative images are shown to the right. Scale: 30 μm. **h, i** Immunoblot detection of p-Smad2/3, total Smad2/3 and GAPDH from human cardiac fibroblasts treated with and without different doses of human EOS lysate (equivalent to $1 \times 10^4$, $2 \times 10^4$, $4 \times 10^4$ EOS/ml) and TGF-β (10 ng/ml) for 30 min. Data are mean ± SEM. The numbers of mice (**c–f**), the numbers of independent experiments (**g–i**), and P values are indicated, one-way ANOVA test (**c, h, i**) and non-parametric Mann-Whitney U test (**d–g**).

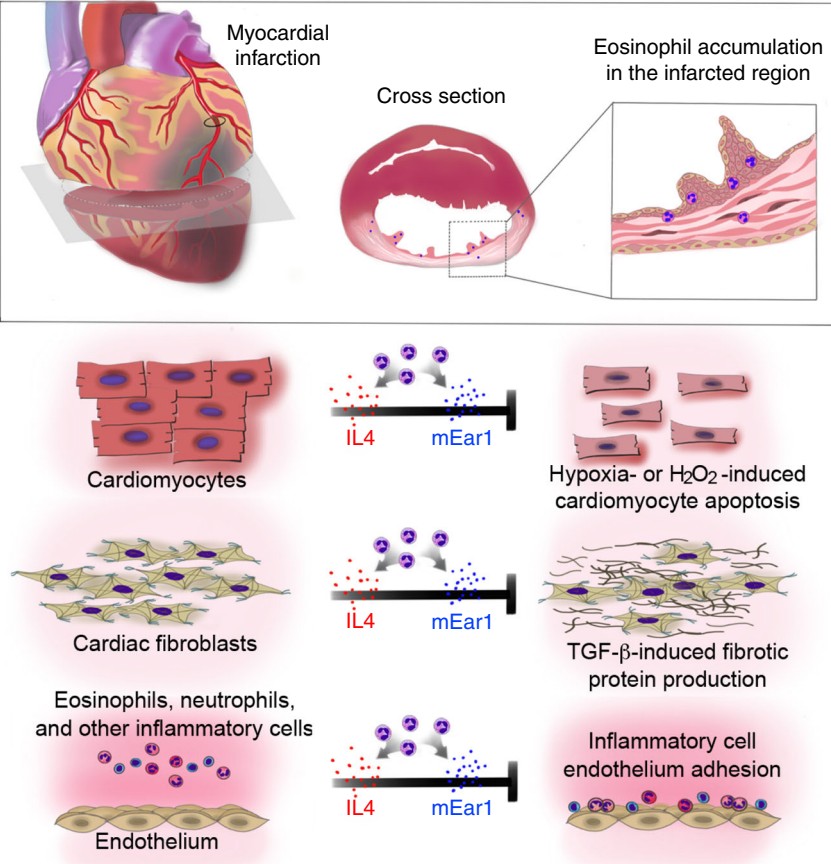

**Fig. 7 EOS functions in post-MI heart. a** Schematic illustration of EOS accumulation in the infarct area of mouse heart post-MI. **b** Summary of EOS function in protecting heart from MI-induced injury by producing Th2 cytokines (e.g., IL4), cationic proteins (mouse mEar1, human ECP and EOS-derived neurotoxin), or other untested molecules to reduce cardiomyocyte apoptosis, cardiac fibroblast activation and fibrotic protein synthesis, and inflammatory cell adhesion and infarct heart accumulation.

water bath. Vortex was performed every 10 min. After digestion, cells were neutralized using 1640 culture medium supplemented with 10% fetal bovine serum (FBS) and washed twice. Cells were then separated using density gradient centrifugation in a 15-ml tube layered with 2 ml 100% Percoll (17-0891-09, Fisher Scientific, Hampton, NH), 1.5 ml 80% Percoll, 1.5 ml 62% Percoll, 1.5 ml 55% Percoll, and 3 ml 45% Percoll that was premixed with the cell preparation. Cells were centrifuged at 800 g for 30 min. To measure EOS, neutrophil, Ly6C$^{hi}$ and Ly6C$^{lo}$ monocytes, dendritic cells (DCs), CD4$^+$ and CD8$^+$ T cells in infarcted heart tissue, heart single-cell preparation was washed and stained with cell viability dye (65-0866-14 and 65-0863-14) and cell surface marker antibodies including CD45 (25-0451-82 and 11-0451-85), CD45.1 (45-0453-82), CD11b (17-0112-82), Siglec-F (12-1702-82), CCR3 (144516, BioLegend, San Diego, CA), Gr-1 (12-5931-82), Ly6C (53-5932-82), CD11c (53-0114-82), MHC-II (107607, BioLegend), CD4 (A15384), and CD8 (12-0081-82) (all from eBioscience, San Diego, CA) to quantify EOS, neutrophils, Ly6C$^{hi}$ and Ly6C$^{lo}$, DCs, CD4$^+$ and CD8$^+$ T cells in the hearts. All cell surface antibodies used here were diluted at 1:400. BD FACSCanto II was used for data collection and Flowjo 10.4.1 was used for data analyses.

**Splenocyte isolation and flow cytometry.** The spleen was removed from the mouse, placed in a cold PBS, and grinded in 5 ml PBS before being filtered through a 70-μm cell strainer. Splenocytes were collected after depleting the red blood cells using the red blood cell lysis buffer (004333). Splenocytes were then stained with cell viability dye and antibodies against CD45, CD11b, Gr-1, CD4, CD8, and CD25 (48-0253-80) (all from eBioscience). To analyze TNF-α$^+$ cells, splenocytes were immunostained with CD4 and CD8 first, then stimulated with a Cell Stimulation Cocktail (00-4975-03, eBioscience) at 37 °C for 5 h in the dark to induce intracellular cytokine expression and accumulation before intracellular staining was performed. Cells were fixed with the IC Fixation Buffer (00-8222-49), permeabilized with the permeabilization buffer (00-8333-56), and immunostained with rat anti-mouse TNF-α-Alexa Fluor 488 (53-7321-82). For the detection of T-regulatory cells, the cells were incubated with anti-mouse CD4 and CD25 antibodies at 4 °C for 30 min in the dark to perform cell surface marker staining.

Cells were then fixed, permeabilized, and stained with the anti-mouse Foxp3 antibody using the Foxp3 Transcription Factor Staining Buffer Kit according to the

manufacturer's instructions (A25864A). All isotype controls were used to ensure antibody specificity under the same conditions. All antibodies and buffers were from eBioscience. All cell surface antibodies used here were diluted at 1:400. TNF-α and Foxp3 were diluted at 1:200. BD FACSCanto II was used for data collection and Flowjo 10.4.1 was used for data analysis.

**Mouse EOS culture.** EOS was cultured as previously reported[52]. In brief, bone marrow cells were collected from mouse femurs and tibias, centrifuged for 5 min at 300 g, followed by lysing red blood cells by pipetting cell pellet up and down twice in 9 ml ddH$_2$O and with immediate subsequent addition of 1 ml 10xPBS. The lysis protocol was repeated for a maximum of 3 times. Once red blood cells were lysed, the cells were suspended in 10 ml of PBS prior to cell counting. Cells with a concentration of 10$^6$ cells/ml were plated in an 80 cm$^2$ flask and cultured in a base media containing mouse stem cell factor (100 ng/ml, 250-03, PeproTech, Rocky Hill, NJ) and mouse Flt-3-ligand (100 ng/ml, 250-31 L, PeproTech) for 2 days. On day 2, one-half of the media from each flask was replaced with a fresh medium containing mouse stem cell factor and Flt-3- ligand. On day 4, culture media were replaced with the same media containing recombinant mouse IL5 (10 ng/ml, 405-ML, R&D Systems, Minneapolis, MN). Half of the culture media was changed with fresh media containing IL5 every two days until the 14$^{th}$ day. On day 10, 12, and 14, cells were collected and cell purity was examined by FACS.

**Mouse heart endothelial cell isolation.** Mouse heart endothelial cells (MHEC) were isolated as previously described[12]. Briefly, mouse hearts were removed from young mice aged between one day and two weeks, washed in 15 mL of cold isolation medium containing DMEM, 20% FBS and 1% penicillin-streptomycin, minced finely with scissors, and finally digested in 25 ml of pre-warmed collagenase B (50 mg into 25 ml of Dulbecco's PBS (DPBS), LS004177, Worthington Biochemical Co) at 37 °C for 45 min. During digestion, vortex was performed every 10 min, followed by the addition of double the volume of isolation medium to stop the digestion. The digested tissue was filtered through a 70 um cell strainer and centrifuged at 400 g for 10 min at 4 °C. The cell pellet was re-suspended in 2 ml of cold DPBS and incubated with PECAM-1 (553389, BD Biosciences, Bedford, MA)-

coated Dynabeads (11035, Thermo Fisher Scientific, Woburn, MA) at a concentration of 15 µl of beads/mL of cell suspension at room temperature for 10 min with end-over-end rotation. The tube containing cells was mounted on a magnetic separator and left for 1-2 min. The supernatant was then removed and cells were re-suspended in 10 ml of growth medium containing Hi Glucose DMEM with 20% FBS, 1% penicillin-streptomycin, 100 µg/mL heparin (H-3933, Sigma-Aldrich), 100 µg/mL endothelial cell growth supplement (ECGS) (BT-203, Sigma-Aldrich), 1x nonessential amino acids, 2 mM L-glutamine, 1x sodium pyruvate, and 25 mM HEPES. Cells with beads were plated in a gelatin-coated T75 flask. About 5~9 days when cells approached and became confluence, the second sort with an identical protocol to the above was performed by using the intercellular adhesion molecule-2 (ICAM-2, 553325, BD Biosciences, Woburn, MA)-coated Dynabeads. In experiments involving EOS or neutrophil adhesion assay, MHECs from passage 2-5 were used.

**Adult mouse cardiomyocytes**. Adult mouse cardiomyocytes were isolated and cultured as previously reported[53,54]. Hearts removed from heparin pre-treated mice were perfused and digested in a perfusion system as described[53], followed by cell dissociation and calcium reintroduction. Cells were then planted in laminin (1-2 µg/cm2)-precoated 6-well plate in a 2% $CO_2$ incubator at 37 °C. Cells were incubated for one to three hours to make the attachment rate about 80%. After attachment, plating medium was gently removed and replaced with culture medium. Adult mouse cardiomyocytes were pretreated with EOS or EOS lysates prepared from WT, $Il4^{-/-}$, $Il10^{-/-}$, $Il13^{-/-}$ mice or rabbit anti-mouse mEar1 polyclonal antibody (orb13385, Biorbyt, San Francisco, CA) for overnight. Cells were then treated with 100 µM $H_2O_2$ or sterile water for four hrs before harvest. Cell apoptosis was detected using the FITC-Annexin V apoptosis detection kit-I (556547, BD Biosciences), followed by FACS analysis. The expression of Bcl2 in mouse cardiomyocytes was detected by immunoblot using rabbit mAb (1:1000, #3498 S, Cell Signaling Technology, Beverly, MA).

**Adult mouse cardiac fibroblasts**. Mouse cardiac fibroblasts were isolated from the cardiomyocyte-fibroblast mixture by centrifugation at a low speed of 20 g. Fibroblasts on the top layer were collected and cultured in high glucose DMEM containing 10% FBS and 1% penicillin-streptomycin. Cells from 2 to 5 passages were used. Fibroblasts at about 70% confluence on a 6-well plate were starved for 24 hrs and then stimulated with or without different mouse EOS lysate at $1 \times 10^6$ EOS/ml or mouse recombinant mEar1 (10, 100, 1000 ng/ml, 00128-03, Aviscera Bioscience Inc, Santa Clara, CA) overnight then treated with or without TGF-β (10 ng/ml, 14-8342-80, eBioscience) for 30 min. Protein extracts from fibroblasts were separated on 10% SDS–PAGE and then transferred to a polyvinylidene fluoride (PVDF) membrane. After blocking with 5% nonfat milk for two hrs, the membrane was incubated with rabbit anti-mouse total Smad2 (1:1000, 5339 S), rabbit anti-mouse p-Smad2 (1:1000, 3108 S), rabbit anti-mouse total Smad3 (1:1000, 9523 S, (all from Cell Signaling Technology), rabbit anti-mouse p-Smad3 (1:1000, ab52903, Abcam, Cambridge, MA), and rabbit anti-mouse GAPDH (1:1000, 2118 S, Cell Signaling Technology) at 4 °C overnight, followed by incubation with an HRP-conjugated secondary antibody (1:3000, G21234, Thermo Fisher Scientific) for two hrs. The resulting signals were detected using the Amersham ECL Prime Western Blotting Detection Reagent (RPN2236, Fisher Scientific, Hampton, NH). GAPDH was used to ensure equal protein loading. Image lab 6.0 was used for immunoblot data analyses.

**Mouse lymphocytes**. Mouse splenic lymphocytes were isolated using density gradient centrifugation with the LSM™ lymphocyte separation medium (0850494X, MP Biomedicals, Solon, OH) according to the manufacturer's instructions, and were stimulated using plate-bound rat anti-mouse CD3 monoclonal antibodies (5 µg/ml, 100223, BioLegend) and hamster anti-mouse CD28 monoclonal antibodies (3 µg/ml, 102121, BioLegend) in 48-well plates ($1 \times 10^6$ cells/well) containing complete RPMI 1640 culture medium supplemented with 10% FBS and 1% penicillin-streptomycin at 37 °C with 5% $CO_2$ to activate T-cell receptors. Recombinant mouse TGF-β1 (10 ng/ml, 14-8342-80, eBioscience) and IL2 (10 ng/ml, 402-ML, R&D system) were used to induce T regulatory cell differentiation. The lymphocytes were treated with or without WT EOS. Three days later, cells were harvested and tested for the percentage of T regulatory cells.

**Human blood EOS isolation**. Human EOS were isolated from human donor blood by negative selection using a commercial EOS isolation kit (130-104-446, Miltenyi Biotec Inc, Somerville, MA) according to the manufacturer's instructions. In brief, the leukocyte rich buffy coat was purchased from Massachusetts General Hospital (Boston, MA). EOS were isolated using density gradient centrifugation in a 50 ml tube layered with 20 ml 75% Percoll (17-0891-09, Fisher Scientific), 10 ml 65% Percoll, and 20 ml blood that was prediluted by adding one volume of PBS. Cells were centrifuged at 800 g for 30 min. The plasma and lymphocytes were carefully removed and discarded and the cell layer containing EOS between 75 and 65% Percoll gradients was collected. Cells were then washed with PBS and autoMACS Pro washing solution (130-092-987, Miltenyi Biotec Inc) twice separately, followed by antibody staining according to the manufacturer's instructions (130-104-446, Miltenyi Biotec Inc). The cells with antibodies were placed in the magnetic field of

the MACSxpress Separator for 15 min. The magnetically labeled cells adhered to the wall of the tube while the supernatant containing EOS was carefully collected in a new 50 mL tube and then washed twice. Cells were collected and counted. The concentration of human EOS lysate was $10^5$ cells/ml in 1640 culture medium. Use of discarded human blood specimens was also approved by the Human Investigation Review Committee at the Brigham and Women's Hospital, Boston, MA, USA (protocol #2010P001930) with a waiver of informed consent since the use of discarded human blood specimens did not involve patient contact or release of patient information.

**Human cardiac myocytes and fibroblasts**. Human cardiac myocytes (HCM, C-12811, PromoCell, Heidelbery, Germany) were isolated from normal human ventricle tissue of the adult heart. To induce terminal differentiation, HCM were plated and cultured in 8- well plates in a myocyte growth medium (C-22070, PromoCell) containing 0.05 ml/ml fetal calf serum (FCS), 0.5 ng/ml recombinant human epidermal growth factor, 2 ng/ml recombinant human basic fibroblast growth factor, and 5 µg/ml recombinant human insulin for 60 days. During this period, culture medium was replaced every week. Then, HCM were treated with or without human EOS lysate at a concentration of $2.5 \times 10^3$, $1 \times 10^4$, $2.5 \times 10^4$ cells /ml in a hypoxia chamber at 37 °C for 36 hrs to induce apoptosis, followed by immunofluorescent TUNEL staining according to the manufacturer's instructions (11684795910, Sigma-Aldrich). HCM from 3 to 5 generations were used. Human cardiac fibroblasts (HCF, C-12377, PromoCell) were isolated from the ventricles of the adult heart, and cultured in 6-well plates in fibroblast growth medium-3 (C-23025, PromoCell) containing 0.1 ml/ml FCS, 1 ng/ml recombinant human basic fibroblast growth factor, and 5 µg/ml recombinant human insulin. To test the function of EOS in SMAD-signaling, HCF were pre-treated with or without human EOS lysate at the concentration of $1 \times 10^4$, $2 \times 10^4$, $4 \times 10^4$ cells /ml at 37 °C with 5% $CO_2$ for 2 hrs, followed by treating cells with or without TGF-β (10 ng/ml) for 30 min. HCF were harvested to test the expression of total Samd-2/3 (1:1000, 5339 S, 1:1000, 9523 S, Cell Signaling Technology) and p-Samd-2/3 (1:1000, 3108 S, Cell Signaling Technology, 1:1000, ab52903, Abcam). HCF from 3 to 5 generations were used. Image lab 6.0 was used for immunoblot data analyses.

**Histological analysis**. To determine the location of EOS in infarct hearts, eoCRE$^{+/-}$GFP$^{+/-}$ mice were used. At 1-day post-MI, 5 µm heart sections were prepared and used for immunofluorescent staining with rabbit anti-mouse GFP antibody (ab183734, 1:500, Abcam). Sections were stained with rabbit anti-mouse cleaved caspase-3 monoclonal antibody (1:100, 8172, Cell Signaling Technology) together with mouse anti-mouse α-SMA monoclonal (1:500, F3777, Sigma-Aldrich) and rabbit anti-mouse cardiac myosin heavy chain (MYH) polyclonal antibody (1:100, bs-15444R-A488, Bioss Inc., Woburn, MA). For the control slides, the primary antibody was replaced with PBS. Terminal deoxynucleotidyl transferase dUTP nick end labeling (TUNEL) staining was used to detect dead cells in the heart tissue from 1-day post-MI according to the manufacturer's instructions (11684795910, Sigma-Aldrich). Haemotoxylin and eosin (H&E) staining was performed to test MI infarct size according to manufacturer's instruction (HT110116, Sigma-Aldrich). Masson's trichrome kit was used to detect heart collagen deposition (87019, Fisher Scientific).

**Neutrophil adhesion assay**. Neutrophils were isolated from murine bone marrow. Briefly, bone marrow cells were collected from mouse femurs and tibias, and centrifuged for 5 min at 300 g. Cells were then separated using density gradient centrifugation in a 15-ml tube layered with 4 ml 75% Percoll (17-0891-09, Fisher Scientific), 4 ml 65% Percoll, and 4 ml 55% Percoll that was pre-mixed with the cell suspension. Cells were centrifuged at 800 g for 30 min at room temperature. Neutrophils were collected from the layers between 75 and 65% Percoll, and washed twice with PBS. FACS examined the purity of neutrophils before use. Mouse heart endothelial cells (MHEC) were planted and cultured on 4-chamber slides for 2-3 days to reach a density of >80%. MHEC were then treated with or without (control) 100 ng/ml TNF-α for 12 hrs, followed by co-culturing for 30 min with $10^6$ EOS/ml from WT, $Il4^{-/-}$, $Il13^{-/-}$, or WT EOS pre-treated for 10 min with mEar1 antibody (1:200, orb13385, Biorbyt). Neutrophils were labeled with 5-(and 6)- carboxyfluorescein diacetate succinimidyl ester (CFSE) according to the manufacturer's instructions (65-0850, eBioscience). MHEC and EOS co-culture was then treated with $2 \times 10^5$/ml CFSE-tracked neutrophils for 1 hr. Culture medium and unattached cells were removed by washing MHEC monolayer with PBS twice. Slides were mounted and captured and CFSE-tracked neutrophils were counted.

**Real-time PCR**. Total RNA from heart samples was prepared using the TRIzol™ reagent (15596018, Fisher Scientific) and cDNA was reverse-transcribed using the high capacity cDNA reverse transcription kit (4368813, Thermo Fisher Scientific). The relative mRNA levels of target genes were quantified using the iTaq UniverSYBR Green SMX 5000 (1725125, Bio-Rad Laboratories Inc. Hercules, CA) with an ABI PRISM 7900 sequence detector system (Applied Biosystems Co, Foster City, CA). Each reaction was performed in duplicate and changes in relative gene expression levels were normalized to β-actin levels using the relative threshold cycle method. Primer sequences are listed in Supplementary Table 4.

**Immunoblot**. Immunoblot determined the expression of IL4 or mEar1 in mouse heart tissue extracts and EOS from WT and *Il4*[−/−] mice. Equal amount of protein (mEar1: 5 µg cell lysate or 20 µg heart tissue lysate. IL4: 60 µg heart tissue lysate) was separated on 12–14% SDS–PAGE and then transferred to polyvinylidene fluoride (PVDF) membranes. Membrane was blocked overnight in 5% bovine serum albumin in Tween (0.1%)-PBS, followed by primary antibodies IL4 (1:500, ab11524, Abcam) and mEar1 (1:1000, orb156688, Biorbyt) at 4 °C overnight. Membrane was striped and reprobed with the GAPDH antibody (1:1000, 2118 S, Cell Signaling Technology) to ensure equal protein loading.

**Statistical analysis**. The EOS counts of attending men with and without previous AMI were compared by Student's t-test, and tested multivariately by logistic regression analysis, while differences in NYHA classification were tested by ANOVA. All mouse data were expressed as mean ± SEM. We used non-parametric Mann–Whitney $U$ test followed by Bonferroni correction to compare two-group data that did not pass the normality test. Non-parametric Kruskal–Wallis test (one-way analysis of variance on ranks) was used for all mouse and cell culture data analysis that contained multiple group comparisons and that did not pass the normality test. SPSS16 version was used for analysis and $P < 0.05$ were considered significant.

**Reporting summary**. Further information on research design is available in the Nature Research Reporting Summary linked to this article.

## Data availability

Data supporting the findings of this manuscript are available from the corresponding author upon reasonable request. The source data underlying Figs. 1a, b, e–g, i, 2c, d, f–i, 3a–g, 4a–e, g, 5c–h, j, k, 6c–i, and Supplementary Figures 1a–f, 2a, b, e, 3a–f, 5c, d, 6c, d, 7c, 8c, 9a, b, 10b–d, 11a–h, 12a, b, e and 13a, b are provided as a Source Data file. Source data are provided with this paper.

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

## Acknowledgements

We thank Ms. Eugenia Shvartz for her technical assistance and Ms. Chelsea Swallom for her editorial assistance. We also thank Dr. Helene Rosenberg from the National Institutes of Health for help with mouse eosinophil culture, Dr. Elizabeth Jacobson from Mayo Clinic Arizona, Scottsdale, AZ for providing the *eoCRE* and iPHIL mice, Dr. Qingen Ke from the Department of Medicine, Beth Israel Deaconess Medical Center for help with the mouse myocardial infarction model, Dr. Sudeshna Fisch from the Cardiovascular Physiology Core, Brigham and Women's Hospital for her assistance in echocardiography, and Dr. Kay W. Case from the Department of Pathology, Brigham and Women's Hospital for her assistance in mouse heart endothelial cell isolation. This work is supported by grants from National Natural Science Foundation of China [91939107 and 81770487 to JG], the Chinese Academy of Medical Sciences Innovation Fund for Medical Sciences [2016-I2M-1-006 to JW, and 2019-I2M-5-023 to JG], the Open program for Key Laboratory of Emergency and Trauma of Ministry of Education Hainan Medical University [KLET-201917 to JG], and the National Institute of Health [HL080472 to PL, HL123568, HL60942, and AG063839 to GPS], and the RRM Charitable Fund to PL. Dr. Jing Liu is supported by the American Heart Association Postdoctoral Fellowship #20POST35210968.

## Author contributions

J.L., C.Y., T. L., and Z.D. performed most of the in vivo mouse model, in vitro cell culture, and relevant analysis. X.Z., J.L., W.F., Q.H., C.L., Y.W., D.Y., and D.L. assisted the in vivo mouse models, immunoblots, and statistical analysis. G.K.S. helped with tissue immunohistology analysis. J.S.L., A.D., and L.M.R. helped with the DANCAVAS patient studies. G.N., and F.W.L. helped with the mouse heart endothelial cell culture. L.L. helped with the mouse cardiomyocyte culture. J.W., J.G., and G.-P.S. participated in experimental design and data interpretation. P.L. and G.-P.S. wrote the article. All authors contributed to the final editing and approval of the manuscript.

## Competing interests

The authors declare no competing interests.
