## [Peer Review File · Nature Communications]

REVIEWERS' COMMENTS:

Reviewer #4 (Remarks to the Author):

This manuscript establishes the protective role of eosinophils after MI. The mouse data are convincing and provide causality. The human data associations are supportive of translatability. The authors did a meticulous job detailing potential mechanisms but adoptive transfer studies, observing reuse of phenotype and downstream actions.

Coming in at a late stage into this review process, I do not think any additional data are warranted and that the response to reviewer 1 was thorough and earnest. It is a pity that R1 bowed out -- my assessment as a replacement is that the response to R1 is thorough and acceptable.

One minor comment: the number of eosinophils is pretty low. The authors are transparent as they show the peak number per heart after MI is 3,000. However I would also recommend to add this somewhere in the discussion. This number is orders of magnitude lower than myeloid cell recruitment, and may be more in the range of lymphocytes (which apparently can influence recovery from MI also).

Reviewer #5 (Remarks to the Author):

I think the responses by the authors are appropriate and address the main concerns.

Reviewer #6 (Remarks to the Author):

All of R3 comments have been addressed.

Point-by-Point Responses

Reviewer #4 (Remarks to the Author):

This manuscript establishes the protective role of eosinophils after MI. The mouse data are convincing and provide causality. The human data associations are supportive of translatability. The authors did a meticulous job detailing potential mechanisms but adoptive transfer studies, observing reuse of phenotype and downstream actions.

Coming in at a late stage into this review process, I do not think any additional data are warranted and that the response to reviewer 1 was thorough and earnest. It is a pity that R1 bowed out -- my assessment as a replacement is that the response to R1 is thorough and acceptable.

One minor comment: the number of eosinophils is pretty low. The authors are transparent as they show the peak number per heart after MI is 3,000. However I would also recommend to add this somewhere in the discussion. This number is orders of magnitude lower than myeloid cell recruitment, and may be more in the range of lymphocytes (which apparently can influence recovery from MI also).

Response:

We thank this reviewer for his/her time and effort spent to evaluate our prior submission. We are happy to hear from this reviewer that our prior revision was thorough and satisfactory. We agree fully with this reviewer that the eosinophil (EOS) count in post-MI heart was low (Figure 1f). Relative to CD11b⁺Ly6G⁺ neutrophils and CD11b⁺Ly6C^{hi} monocytes, EOS counts were on the orders of magnitude lower. However, heart EOS counts were comparable to those of CD4⁺ and CD8⁺ T cells at 1, 3, and 7 days post-MI based on our FACS analysis presented in Figure 3a. Results from Figure 4g indicate a role of EOS in blocking neutrophil adhesion. As this reviewer commented, both pro-inflammatory and reparative inflammatory cells in post-MI heart may also affect EOS accumulation. This study did not test this hypothesis. We discussed this study limitation in our revised Discussion on page 9, lines 38-45.

Page 9, lines 38-45:

Fourth, EOS accumulation in heart post-MI was relatively low compared with many other inflammatory cells. At 1 day post-MI, there were about 3,000 EOS per heart, many fewer than neutrophils and Ly6C^{hi} and Ly6C^{lo} monocytes. At 3 days post-MI, heart EOS counts reached to the level of neutrophils, but still about one third of Ly6C^{hi} monocytes. Results from this study suggest that EOS in heart post-MI affect the accumulation of other inflammatory cells. It is also possible that the accumulation of pro-inflammatory and reparative inflammatory cells control EOS accumulation, a hypothesis remains untested.

Reviewer #5 (Remarks to the Author):

I think the responses by the authors are appropriate and address the main concerns.

Reviewer #6 (Remarks to the Author):

All of R3 comments have been addressed.

Response:

We thank both Reviewers 5 and 6 for their time and effort to evaluate our prior submission and we are pleased to hear that our prior responses and revised manuscript addressed all reviewers' critiques satisfactorily.